# Single molecule imaging simulations with advanced fluorophore photophysics

Dominique Bourgeois [1] [✉]

Advanced fluorescence imaging techniques such as single-molecule localization microscopy (SMLM) fundamentally rely on the photophysical behavior of the employed fluorophores. This behavior is generally complex and impacts data quality in a subtle manner. A simulation software named Single-Molecule Imaging Simulator (SMIS) is introduced that simulates a widefield microscope and incorporates fluorophores with their spectral and photophysical properties. With SMIS, data collection schemes combining 3D, multicolor, single-particle-tracking or quantitative SMLM can be implemented. The influence of advanced fluorophore characteristics, imaging conditions, and environmental parameters can be evaluated, facilitating the design of real experiments and their proper interpretation.

[1] Institut de Biologie Structurale, Univ. Grenoble Alpes, CNRS, CEA, IBS, 38044 Grenoble, France. ✉email: dominique.bourgeois@ibs.fr

Single molecule localization microscopy (SMLM) has become a central tool to decipher the mechanisms of life at the nanoscale[1]. It is now well established that the successful application of SMLM techniques such as PhotoActivated Localization Microscopy (PALM) or direct Stochastic Optical Reconstruction Microscopy (dSTORM) relies on the proper photophysical behavior of the employed fluorophores. Taking this behavior into consideration becomes crucial when quantitative information is sought, for example, in the case of single-particle-tracking[2] or oligomer counting experiments[1]. In general, results provided by many advanced fluorescence imaging methods may be severely biased by poorly controlled fluorophore photophysical properties.

Several simulation packages have been developed in the last years to aid in setting up and interpreting SMLM experiments[3–8]. These packages are very useful to reproduce some of the artifacts (e.g., linkage or clustering artifacts) commonly observed in SMLM. However, they use only relatively simplistic descriptions of fluorophore photophysics, generally based on 3- or 4-state models (typically an inactive state, an active state, a dark state and a bleached state). Furthermore, they do not incorporate spectral characteristics of the fluorophores. Thus, existing SMLM simulators do not allow to thoroughly investigate how a fluorophore responds to specific imaging or environmental parameters. Recently the "Virtual SMLM" software has been introduced, which partly addresses this problem by incorporating laser-sensitive phototransformation rates[9]. Yet, the photophysical complexity of the fluorophores and their spectral response to light is not taken into account so that crosstalk effects or e.g., pH- or triplet-state-dependent mechanisms[10,11] cannot be addressed. More generally, biologists seeking to predict data quality using specific fluorophores and imaging conditions, chemists interested in the design of new dyes or studying their photophysical behavior, and physicists aiming at improved data acquisition schemes would benefit from the possibility to simulate experiments in silico and confront the outcome with real data. A tool to do so has been lacking thus far.

Here, the Single Molecule Imaging Simulator (SMIS) package is introduced to simulate a wide variety of SMLM experiments with advanced photophysical and spectral properties of the fluorophores. Central to SMIS is the computation of the photophysical state of each fluorophore with sub-frame time-resolution in response to any data acquisition scheme specified by the user, typically including multiple lasers. Evolution of the photophysical state is based on a thorough description of fluorophores that includes spectral data of photoactive states and photo- or thermally-induced transformations between all states. Beyond the description of fluorophore photophysics, SMIS incorporates a virtual widefield microscope, able to operate in multicolor and 3D modes. Single-particle tracking of fluorophores exhibiting multiple diffusion states, exchange between those states and confined or directed diffusion in 3D is also available. To highlight the potential of SMIS, five different applications are presented below, where fluorophore photophysics or spectral characteristics are shown to influence SMLM data in a complex manner. The benefits and current limitation of SMIS are presented in the Discussion section. Its practical implementation is described in the Methods section.

## Results

### Optimizing imaging of the nuclear pore complex with mEos4b.
The first application aims at investigating how green-state photophysics of the green-to-red photoconvertible fluorescent protein (FP) mEos4b may affect the quality of PALM images. Two-dimensional image stacks were generated using a Nuclear Pore Complex (NPC) model in which the 32 Nup96 nucleoporins are fused to mEos4b molecules, assuming a FP maturation level of 90%. A mEos4b photophysical model comprising 10 states is employed[12] (Fig. 1a and Supplementary Table S1), with each of the fluorescent green or red states (anionic chromophore) being in rapid equilibrium with their nonfluorescent (neutral chromophore) counterparts. Samples consisting of 100 NPCs were submitted to data acquisition protocols (Supplementary Table S2) similar to that recently described[13]. An example of ground-truth photophysical data (e.g., histograms of on-times, off-times, number of blinks) generated by the SMIS data-analysis tool is shown in Supplementary Fig. S1. The retrieved SMLM images upon processing the SMIS-generated image stacks are shown on Fig. 1b. Correlations between those images and a ground-truth image assuming 100% labeling efficiency of Nup96 are shown in Supplementary Figs. S2 and 1d. In comparison to a case where green-state photophysics is deliberately omitted (no switching/blinking nor bleaching) (Fig. 1b, Panel 1), the quality of the image obtained at 3.5 kW/cm² 561 nm laser power density is reduced when green-state photophysics is considered (Fig. 1b, Panel 2). A number of Nup96 sites are clearly missing, similar to what was observed experimentally with the protein mMaple[13]. Thus, SMIS suggests that the non-optimal effective labeling efficiency of NPCs is mostly a result of green-state photobleaching, in line with previous in vitro experimental work[14]. Of note, Urbach tail effects (anti-Stokes absorption)[14] are predicted by SMIS and it is seen that, at 2 W/cm² 405 nm activation laser power density, ~10% of green-to-red photoconversion events resulted from readout photoconversion induced by the 3.5 kW/cm² 561 nm laser (Supplementary Fig. S3). An interesting observation was that reducing the 561 nm laser power-density to 0.5 kW/cm² produced a degraded image (Fig. 1b, Panel 3). This was contrary to expectations based on ensemble-level calculations (Fig. 1c), which suggested a superior photoconversion efficiency at low readout laser intensity (Fig. 1c, f). SMIS allowed to understand that, despite the better photoconversion efficiency, the lower quality of the image originated from both reduced localization precision (Fig. 1e) and longer on-times of the mEos4b molecules. These longer on times induced a higher localization density throughout data collection, resulting in more mislocalizations due to spot overlaps (Supplementary Fig. S4b). In contrast, the superior quality of the NPC image obtained at 3.5 kW/cm² (Fig. 1b, Panel 2), despite only ~60% photoconversion efficiency, results from the fact that 4 Nup96 nucleoporins are present per NPC corner, reducing the likelihood of fully missing corners in the super-resolution images. Because SMIS takes into account the equilibrium between protonated and anionic states of the mEos4b chromophore in both its green and red states, it is also possible to predict the influence of pH on the retrieved NPC image quality. The SMIS data suggested a subtle interplay between sample pH and chosen illumination conditions, with a lower pH (6.5) being more favorable at higher (3.5 kW/cm²) readout laser intensity and a higher pH (8.5) being more favorable at lower (0.5 kW/cm²) readout laser intensity (Fig. 1b, Fig.1d–f, and Supplementary Note 1). Finally SMIS allowed investigating the interest of using Fermi profiles[15] of the 405 nm laser to better distribute localizations throughout data collection. The simulations suggested that a constant localization density cannot be obtained using Fermi profiles due to mEos4b green-state photophysics (Supplementary Fig. S5). Again, a complex interplay was found between Fermi profile parameters and readout laser intensity (Supplementary Note 1 and Supplementary Figs. S3–S5). The best NPC image was predicted at 3.5 kW/cm² using a tight Fermi profile with reduced data collection time. Fermi profiles also resulted in substantially broader distributions of off-times in the mEos4b red state (Supplementary Fig. S4e), as expected in view of the fact that mEos4b long-lived dark states are light-sensitive.

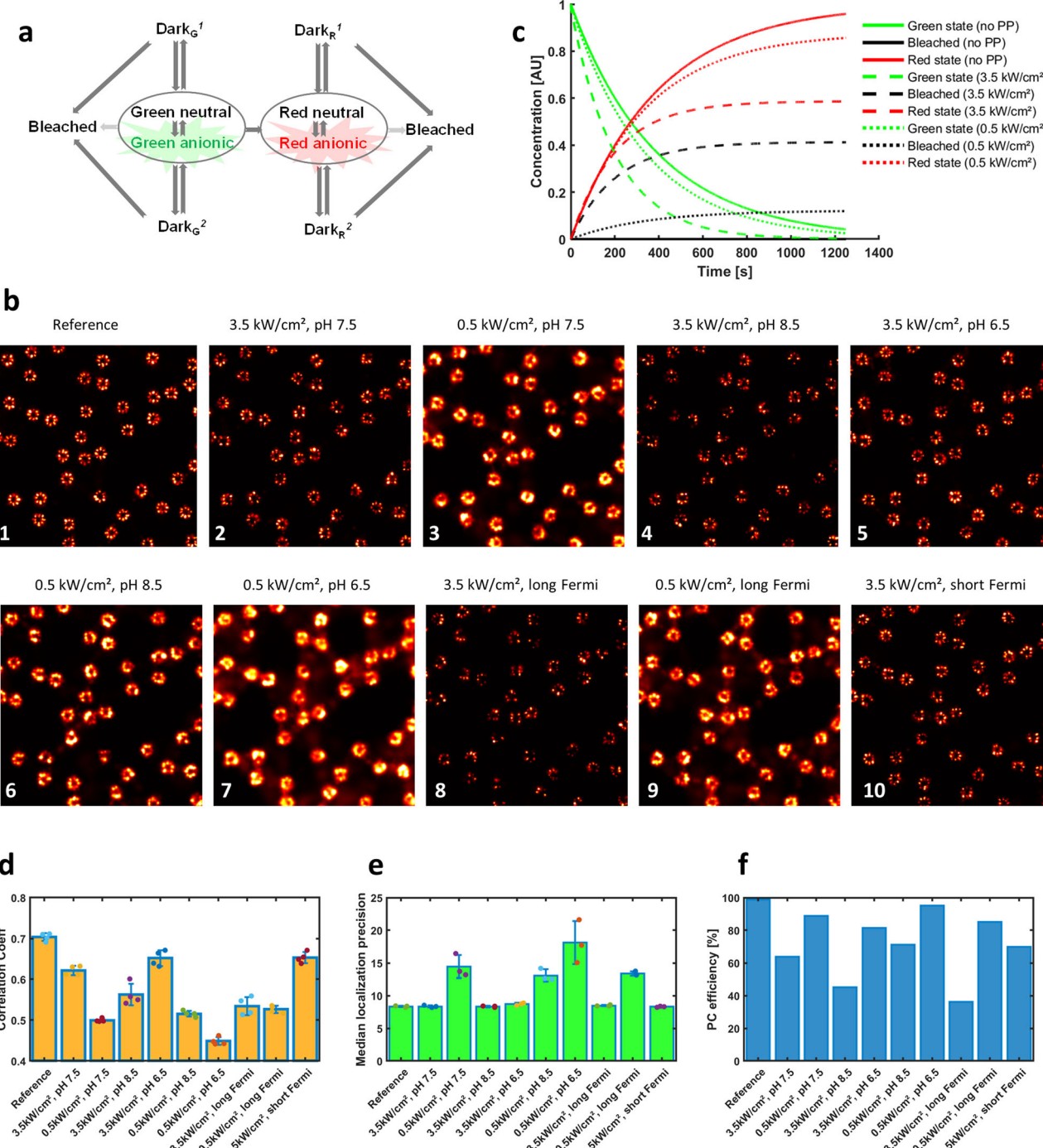

**Fig. 1 PALM SMIS simulations of mEos4b-labeled Nup96 in the Nuclear Pore Complex. a** Employed photophysical model of mEos4b. For both green and red mEos4b, the fluorescent state (anionic chromophore) is in rapid equilibrium with a nonfluorescent state (neutral chromophore). Each colored state can transition to two reversible dark states and one nonreversible bleached state. Dark states may also transition to bleached states. **b** Rendered NPC images (normalized Gaussian) obtained upon processing of SMIS-generated data in various scenarios. pH = 7.5 unless indicated otherwise. Detailed parameters are given in Supplementary Table S2. Green-state photophysics is not considered in the simulation of Panel 1, shown as a reference case. **c** Predicted green-to-red photoconversion efficiency and green-state photobleaching as a function of readout laser intensity. The decay of the green-state population and the rise of the red-state and bleached populations are shown in green, red and black, respectively. Curves are shown when green-state photophysics (PP) is neglected (plain lines), or when it is considered with 3.5 kW/cm² (dashed lines) or 0.5 kW/cm² (dotted lines) readout laser power density, respectively. **d** Correlation coefficients between rendered images and a ground-truth NPC image with 100% labeling efficiency. **e** Median merged localization precision retrieved upon processing the SMIS-generated image stacks. Error bars were computed by randomly splitting the dataset in $n = 3$ subsets and calculating standard deviations between the subsets. Colored dots show individual data points. **f** Ground-truth photoconversion efficiency.

The example above shows that SMIS simulations have the potential to test complex SMLM data collection schemes and possibly hint at optimal parameters. However, the conclusions drawn are only valid as long as the input photophysical models of the used fluorophores are correct. SMIS thus provides a framework to evaluate these conclusions experimentally. Experimental data that would deviate from SMIS predictions could be used to refine the employed photophysical model, serving to increase knowledge on fluorescent protein photophysics.

**Two-color PALM experiments using primed photoconversion**. In the second application, SMIS was used to simulate pseudo two-color PALM experiments based on the so-called "primed-photoconversion" mechanism[10,16,17]. Primed-photoconversion of the fluorescent protein Dendra2 was combined with regular photo-conversion of either mEos4b or PAmCherry[10,16,17]. In such experiments, primed-photoconversion of Dendra2 is first obtained by simultaneous illumination with "priming" light at 488 nm and red-shifted light at 730 nm, referred to as the "priming phase". Regular photoconversion of mEos4b or PAm-Cherry is subsequently obtained with 405 nm laser light, referred to as the "UV-phase". The photoconversion priming mechanism in Dendra2 and other engineered priming variants[18] was proposed to involve excitation of the anionic green-state chromophore at 488 nm, followed by intersystem crossing to a long-lived triplet state[16] which would then be excited by far-red light to finally generate the red-emitting chromophore by a yet unknown mechanism. Although two-color PALM imaging based on primed photoconversion was recently demonstrated to be successful experimentally[16,17], potential caveats such as premature photo-bleaching or residual photoconversion of the non-priming partner by 488-nm light were not extensively studied. Evaluating these caveats can be attempted with SMIS, as shown in Fig. 2. Here, a photophysical model of Dendra2 comprising 12 states was employed (Fig. 2a and Supplementary Table S3)[19], including 2 triplet states accessible from the green and the red chromophore. The virtual sample was based on a neuronal cell, randomly decorated with 5000 of each fluorescent protein (Fig. 2c and Supplementary Fig. S6). The FPs were assumed to exhibit a maturation efficiency of 90%. The FP-labeled targets were set to colocalize within a 5 nm radius. The employed SMIS data collection parameters are shown in Supplementary Table S4. An exemplary trace of a Dendra2 molecule experiencing primed photoconversion is shown in Fig. 2b, demonstrating the complex photophysical trajectory typically followed until photobleaching in the red state. With the chosen photophysical parameters and experimental set up, it can be seen that, as compared to the ground-truth, the mean Coordinate-Based Colocalization (CBC) index[20] and the Nearest Neighbor Distance (NND) retrieved upon processing of the SMIS-generated data are degraded (Fig. 2e and Supplementary Fig. S7). This is corroborated by visual inspection of colocalization, which in the case of the Dendra2/mEos4b pair is clear but not optimal (Fig. 2d, see also Supplementary Fig. S6). Fig 2f suggests that the origin of non-optimal colocalization mostly lies in incomplete photoconversion efficiency due to photobleaching in the green state, with a slight contribution from crosstalk errors due to readout photoconversion of the non-priming partner during the priming phase. Further comparison between the photophysical behaviors of Dendra2, mEos4b, and PAmCherry in the described experiment are discussed in Supplementary Note 2. The study suggests that dark-state shelving in green mEos4b is important to avoid substantial readout photoconversion and massive photobleaching during the priming phase (Fig. 2e, f, Supplementary Fig. S6–S8). SMIS is also able to predict how triplet state saturation may induce differential losses in the effective brightness of mEos4b and Dendra2 in their red states (Supplementary Fig. S9).

The example above shows that SMIS is potentially useful to investigate how fluorophore photophysics affect data quality in multicolor SMLM experiments. The magnitude of the reported effects depends on the input photophysical models, and would need to be established experimentally. In particular, whereas mEos- and Dendra2-variants have been carefully characterized[14,19,21], quantitative knowledge of PAmCherry photophysics is not as thorough. Nevertheless, SMIS can provide a useful qualitative understanding of the interplay between fluorophore characteristics and imaging parameters in such experiments.

**Investigating Cy5 photophysics**. The third application deals with the complex photophysics of cyanine dyes commonly employed in dSTORM experiments[22]. The goal is to investigate whether a recently proposed photocycle of Cy5[23] properly accounts for the experimental dependence on laser power-density of dSTORM data acquired with the closely related dye Alexa647[11]. Based on titration experiments, a nonlinear photobleaching effect of Alexa647 was reported, resulting in a reduced effective labeling efficiency of NPC-Nup96 when the level of red light was raised[11]. Here, the photophysical model of Cy5, described in Fig. 3a, comprises the fluorescent state, the μs-lived triplet state T1, the long-lived sulfur-adduct RS$^-$ (the main dark state in dSTORM), as well as ms-lived radical anionic and cationic states that can be accessed through T1. All these states can photobleach or inter-convert through light-induced or thermal processes as reported by Gidi et al.[23]. The *cis*-state was however not included in the scheme, as *cis-trans* isomerization is a very rapid process that can be considered as a non-radiative relaxation pathway lowering the fluorescence quantum yield of Cy5. Absorption spectra, inter-conversion thermal rates and quantum yields were taken from Gidi et al.[23] (Supplementary Tables S5–8 and Supplementary Fig. S10). The buffer conditions used in both studies[11,23] were essentially identical (no oxygen, β−mercaptoethanol concentra-tion = 143 mM, pH 8), which is important because photo-transformations in organic dyes are highly dependent on environmental conditions. Here, NPCs were simulated in 3D using an astigmatic point spread function. A three-dimensional view of an NPC reconstructed from the simulated data is shown in Fig. 3b. A first titration experiment was performed with SMIS (Supplementary Table S9), and representative rendered images collected at 6.4 and 480 kW/cm² of 647 nm light are shown in Fig. 3c. These images are in line with the expected lower quality of the reconstruction at high power density. Photobleaching quan-tum yields from the fluorescent and the triplet state (which, to the author's knowledge, have not been reported experimentally) were adjusted to quantitatively match the experimentally observed labeling efficiency of dSTORM data at the two laser power den-sities, providing values of $\sim 5 \times 10^{-7}$ and $\sim 5 \times 10^{-3}$, respectively. An extremely high photobleaching quantum yield of the triplet state was thus required to reproduce the experimental data. Fig 3d shows the evolution of the photophysical state's populations during the first 1000 frames of the two simulations (discarded in the reconstructed dSTORM images[11]), using the ensemble simulation tool of SMIS. At 480 kW/cm² of 647 nm laser light, the populations of the triplet state, and to a lower extent of both the anionic and cationic radical states quickly exceed that of the fluorescent state. As a consequence, intense photobleaching occurs at the early stage of data collection, affecting close to ~60% of the Alexa647 population after 1000 frames, to be compared with ~20% at 6.4 kW/cm² (Supplementary Fig. S11). Yet, the single-molecule simulated data (Fig. 3e, f, light-color plain lines) are only partially in line with the experimental data of ref. [11]

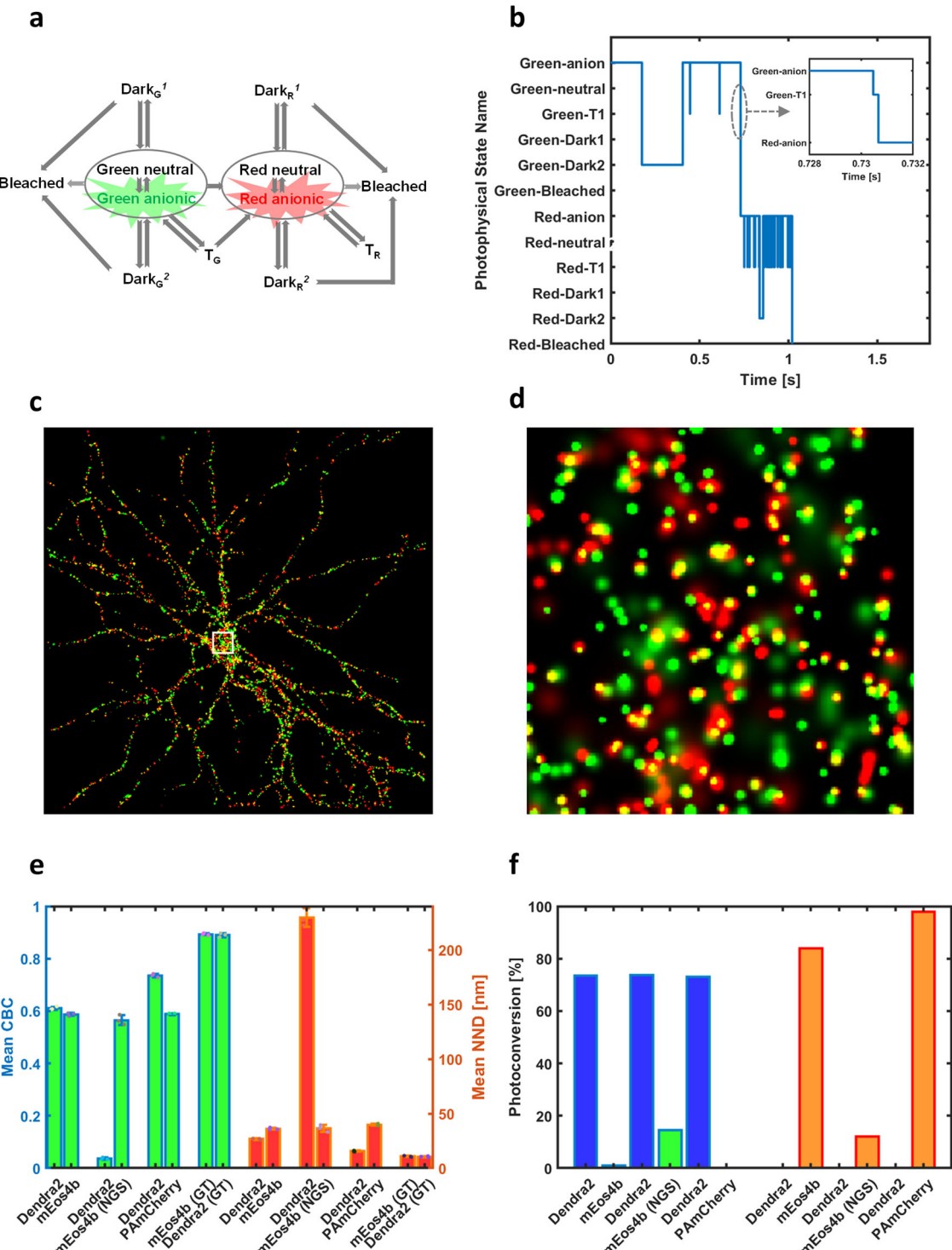

**Fig. 2 SMIS simulations of pseudo 2-color PALM imaging using primed photoconversion. a** Employed photophysical model of Dendra2. In addition to the photophysical states described for mEos4b (Fig. 1a), triplet states accessible from both the green and red states are considered. **b** Example of a Dendra2 time trace showing the sequentially visited photophysical states. The inset shows that conversion to the red state goes via the triplet state through primed photoconversion. **c** Rendered 2 color image (green: mEos4b, red: Dendra2) following priming phase and UV-phase illumination schemes. The area within the white square is enlarged in (**d**) for inspection of colocalization events (yellow). **e** Mean Coordinate Based Colocalization (CBC) index and mean Nearest Neighbor Distance (NND) for the various fluorescent protein pairs tested. Error bars show standard deviation for $n = 3$ simulations. Colored dots show individual data points. **f** Ground truth photoconversion efficiency of the different fluorescent proteins tested during priming phase (left) or UV phase (right). NGS: no green state photophysics considered; GT: ground truth.

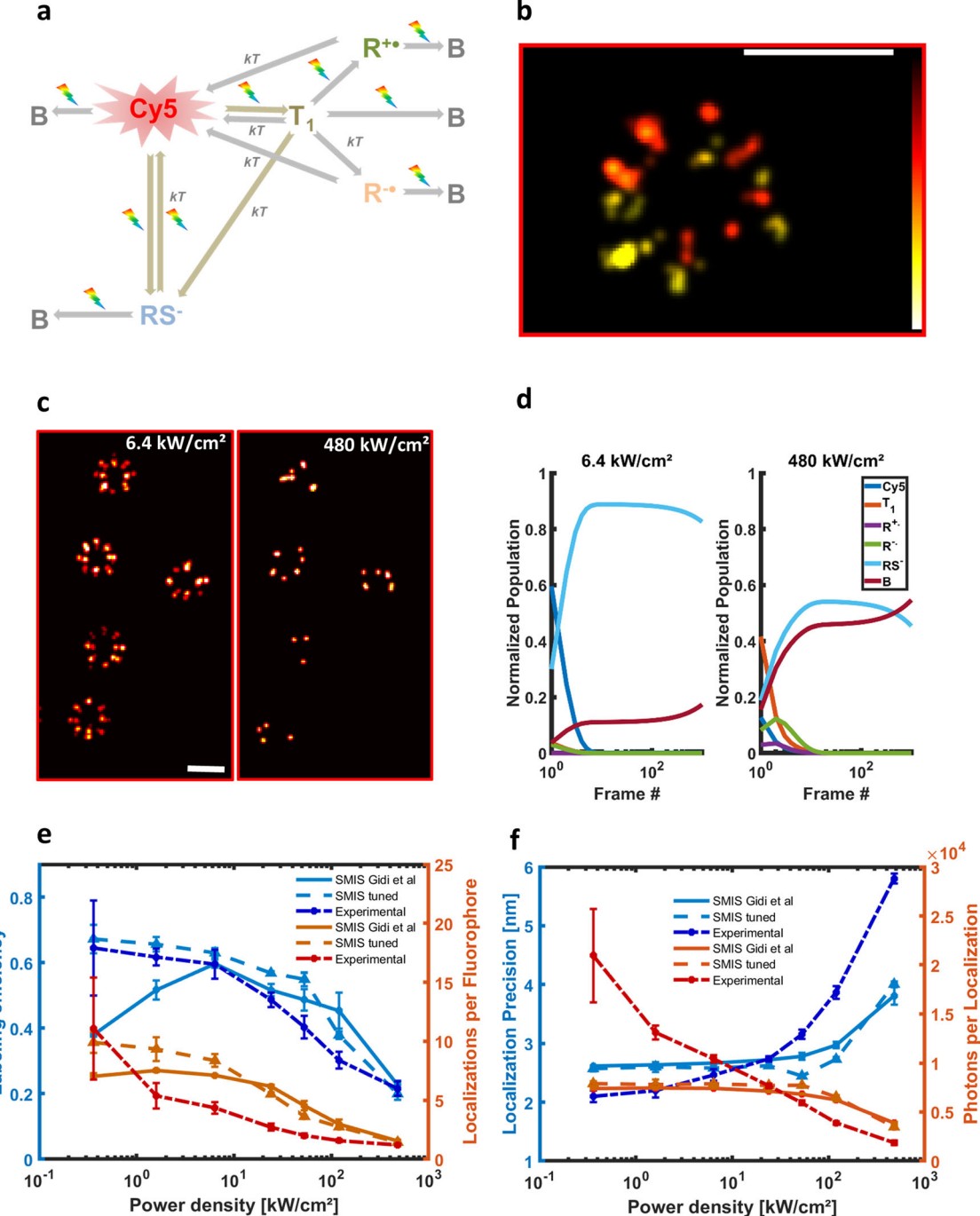

**Fig. 3 dSTORM SMIS simulations of the Nuclear Pore Complex to investigate Cy5 photophysics. a** Employed photophysical model of Cy5. $T_1$: triplet state; B: bleached state; $RS^-$: sulfur anion adduct; $R^{+\bullet}$: radical cation; $R^{-\bullet}$: radical anion; kT: thermal relaxation; rainbow arrow: light sensitivity. **b** 3D view of a reconstructed NPC (color map encodes position along optical axis, scale bar: 100 nm). **c** Rendered NPC's using 6.4 kW/cm² (left) or 480 kW/cm² (right) 647 nm laser power density. 2D projections are shown. Scale bar: 100 nm. **d** Ensemble SMIS simulations showing the evolution of fractional populations of all photophysical states at 6.4 kW/cm² (left) or 480 kW/cm² (right) 647 nm laser power density. **e** Measured labeling efficiency (SMIS data, blue; experimental data, dark blue), and number of localizations per fluorophore (SMIS data, orange; experimental data, red), as a function of the used 647 nm laser power density. Experimental data were taken from ref. [11]. Error bars for SMIS data show standard deviation for $n = 3$ simulations. Individual data points not shown for clarity. Plain lines: results using photophysical values of ref. [23]; Dashed lines: refined photophysical values. **f** Measured localization precision and number of photons per localization. Color and line coding identical to (**e**).

(Fig. 3e, f, dark color dashed-dotted lines) (Supplementary Note 3 and Supplementary Fig. S12). Aiming at a better matching of the simulated and experimental data, SMIS was used to refine rates in the Cy5 photophysical model of ref. [23]. This is justified by the facts that (i) Alexa647 and Cy5 slightly differ chemically, (ii)

Alexa647 in Diekmann et al.[11] was bound to SnapTag whereas Cy5 in Gidi et al.[23] was free in solution, and (iii) the cellular environment in Diekmann et al. might also play a role. The refined model improved the agreement with experimental data, but only partly (Fig. 3e, f, light-color dashed lines, Supplementary

Fig. S12). For example, the experimentally observed high labeling efficiency at low power density was correctly retrieved, but the very high number of photons per localization experimentally observed at low power density could not be reproduced. We reasoned (Supplementary Note 3, Supplementary Note 4) that the remaining discrepancies likely originate from the fact that, in addition to T1, a long-lived (ms) dark state yet to be identified may be centrally involved in controlling Cy5 photobleaching under dSTORM conditions. Interestingly, hints towards the existence of such a state has been provided experimentally[24,25].

The application above shows that SMIS can be a powerful tool to assess and refine the subtle photophysical behavior of fluorescent dyes, and even guide future mechanistic investigations. In particular, the present work may trigger experimental efforts to question the existence of an additional long-lived dark state in the Cy5 photocycle.

**Multicolor imaging with spectral unmixing**. The fourth application deals with spectral unmixing[26,27] for simultaneous multicolor SMLM. It aims to demonstrate how spectral characteristics of fluorophores can be used in SMIS simulations. A 3-color dSTORM experiment was simulated, inspired from the recent study of Andronov et al.[28] that used the fluorophores Alexa647, CF660C, and CF680. A 2D virtual sample was assumed, composed of microtubules, actin fibers, and clathrin-coated pits labeled with Alexa647, CF660C, and CF680, respectively. We did not specifically focus on the complex photophysics of these bright cyanine dyes, but instead used a common simplified "ad-hoc" photophysical model depicted in Fig. 4a and Supplementary Table S10. This model incorporates the long-lived sulfur-adduct and the presumed ms-lived dark state evoked in the previous section of the paper, which is sufficient to reproduce the expected blinking behavior in dSTORM conditions. In this SMIS simulation, we considered split detection on two independent EMCCD cameras, based on real dichroic and emission filters (Supplementary Table S11). The excitation and emission spectra of the 3 fluorophores, as well as the transmission spectrum of the dichroic filter are shown in Fig. 4b. Data collection parameters were similar to those in Andronov et al.[28] (Supplementary Table S11). The simulated data were then processed using SplitVisu[28]. The univariate histogram of the ratios of intensities measured in the two detection channels is presented in Fig. 4c, in comparison with the ground truth for which ratios are only affected by Poisson statistics and camera noise. In this histogram, which closely resemble the experimental one[28], it is observed that the peaks from the processed SMIS data are much broader than those from the ground truth localizations. Nevertheless, based on extracting color channels from a bivariate histogram as performed with SplitVisu[28] (Supplementary Fig. S13a), a rendered image of excellent quality was obtained (Fig. 4d), with only few crosstalk events between the fluorophores. The levels of crosstalk between pairs of fluorophores, not surprisingly, were found to essentially depend on the spectral shift between the fluorophores. They reached a maximum of ~2% under the simulated conditions, as confirmed by SMIS control experiments carried out with single fluorophores (Supplementary Fig. S13). Next, SMIS was used to interrogate which imaging parameters may critically influence the resulting crosstalk level, beyond the photophysical properties of the fluorophores. To this aim, 2-color simulations were performed with Alexa647 and CF660C labeling microtubules and clathrin, respectively, while changing the intensity of the 405 nm laser or the level of fluorescence background. The results, presented in Fig. 4f and Supplementary Fig. S14 clearly indicate that the localization density (affected by the 405 nm laser intensity) did not noticeably influence crosstalk levels, while background

fluorescence had a drastic effect. This derives from the fact that most overlapped localizations were rejected before unmixing, while a high level of background decreased the precision of spot intensity measurement during localization, strongly affecting the retrieved intensity ratios between channels. Thus, the broader univariate histogram as compared to ground truth in Fig. 4c is essentially a consequence of errors in spot intensity evaluation by localization software. However, it is important to note that the overall quality of the super-resolution images did not necessarily correlate with the level of crosstalk, as emphasized in the case of high localization density, clearly resulting in a degraded image (Supplementary Fig. S14). Finally, we interrogated the effect of fluorophore photophysics on crosstalk, by simulating a 2-color dataset employing Alexa647 and Alexa700, as previously described in Lampe et al.[29]. Despite the large spectral shift between the 2 fluorophores, it could clearly be seen that Alexa700 leaked into the Alexa647 channel substantially more than CF680, while producing a lower quality super-resolution image (Supplementary Fig. S15). This is interpreted as a direct consequence of AF700 being dim, not optimally excited by a 642 nm laser, and in general poorly suitable for dSTORM imaging due to a reduced duty cycle.

The example above shows that SMIS simulations allow exploring how spectral characteristics of fluorophores may influence the outcome of multicolor SMLM data acquisition schemes. Beyond the effects of the two parameters studied here (background fluorescence and localization density), the influence of many more parameters could be tested, such as for example, the consequences of modifying optical filters, power and wavelength of the excitation laser, or the photophysical properties of the employed fluorophores.

**Single particle tracking of nucleoid-associated proteins in bacteria**. The fifth application involves single-particle-tracking PALM (sptPALM)[30]. It seeks to investigate how exchange kinetics and molecular confinement combined with fluorophore photophysics may influence accurate retrieval of diffusion coefficients and exchange rates between populations of diffusing molecules. The scenario depicted in Fig. 5a was considered, inspired from the recent study by Stracy et al.[31] of Nucleoid Associated Proteins (NAPs) diffusion dynamics in E. coli. Here, NAPs were assumed to slowly exchange between two main states: a tightly DNA-bound state and a searching state. The searching state is itself a combination of two rapidly exchanging sub-states: a freely-diffusing state and a weakly DNA-bound state. SptPALM data were generated with SMIS considering a virtual 3D sample made of 100 randomly oriented bacteria (Fig. 5b and Supplementary Fig. S16a). The bacteria were assumed to generate slowly bleaching fluorescence background (Supplementary Fig. S16b). The NAPS were labeled with either mEos3.2 or PA-JF549[32], assuming that this photoactivatable dye can be successfully targeted to the bacterial nucleoid to bind e.g., a HaloTag-NAP construct. The photophysical properties of PA-JF549 were derived from ref. [32] (Supplementary Tables S12, 13). The study was also aimed at evaluating the potential benefit of using weak 488 nm light in addition to standard 561 and 405 nm illumination to increase the tracklengths of mEos3.2-labeled molecules[33]. Data collection parameters were set essentially as in Stracy et al.[31] (Supplementary Table S14), except that a stroboscopic mode was employed to reduce temporal averaging[34] (Supplementary Fig. S16c): the frametime was 15 ms, but illumination was switched on only for 5 ms in each frame. Care was taken in adjusting the activation light (405 nm) to achieve stringent localization sparsity, ie ~80% of the tracks not overlapping in time with other tracks in single nucleoids. Using Trackit[35] and UTrack[36] the expected difference in average tracklength between the 2 types of

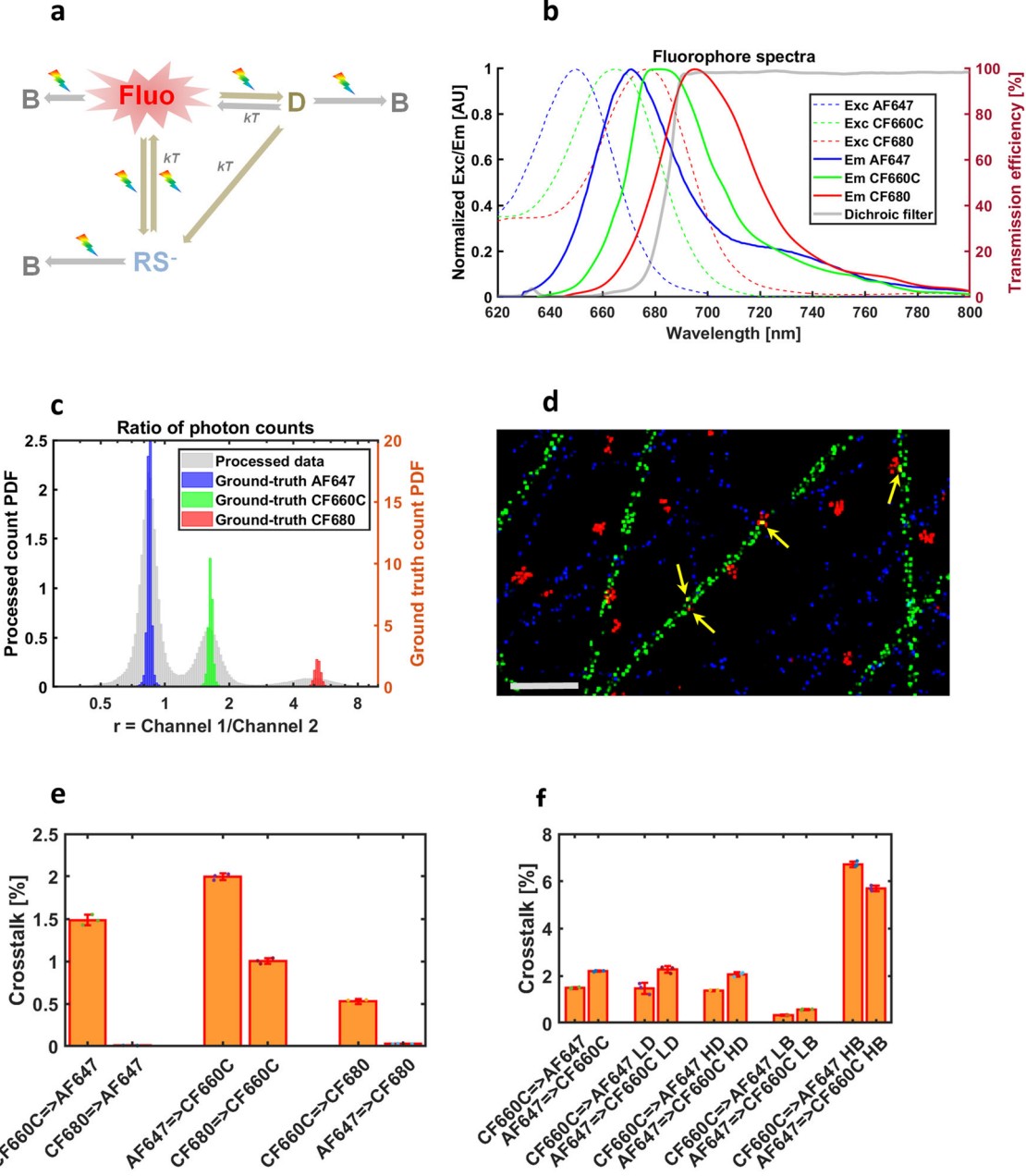

**Fig. 4 Spectral demixing SMIS simulations of Alexa647, CF660C, and CF680 labeling microtubules, actin and clathrin. a** Employed photophysical model for the three fluorophores. D: ms-lived dark state; B: bleached state; RS⁻: sulfur anion adduct; kT: thermal relaxation; rainbow arrow: light sensitivity (**b**) excitation (dashed lines) and emission (plain lines) spectra of AF647 (blue), CF660C (green) and CF680 (red) as used in SMIS. The transmission spectrum of the dichroic filter is shown in gray (right axis). **c** Univariate histogram of fluorophore count ratios in SMIS-generated 3-color spectral demixing dataset. The histogram from processed data (left axis) is overlaid on the ground truth histograms for the 3 fluorophores (blue: AF647; green: CF660C; red: CF680) (right axis). r: ratio of photon counts between channel 1 (shorter wavelengths) and channel 2 (longer wavelengths). **d** Extract of the rendered unmixed 3-color image (blue: AF647; green: CF660C; red: CF680). Examples of crosstalk events are highlighted by yellow arrows. Scale bar: 1 µm. **e** Crosstalks between pairs of fluorophores in the unmixed 3-color image. **f** Crosstalk between AF647 and CF660C in unmixed 2-color images acquired under various conditions (LD/HD: Low/High localization density; LB/HB: Low/High background fluorescence). Error bars in (**e**) and (**f**) were computed by randomly splitting datasets in $n = 3$ subsets. Colored dots show individual data points.

fluorophores was observed (Supplementary Fig. S16d). However it could be seen that the fidelity of track reconstruction only reached about 80% (Supplementary Fig. S17), which is mostly assigned to the difficulty in disentangling crossing tracks within a highly-confined volume. Next, apparent individual diffusion coefficients based on mean jump distances from tracks containing at least 5 localizations were calculated as described in Stracy et al.[31], and histograms were produced (Fig. 5c) from which

fractional populations and overall apparent diffusion coefficients of the tightly DNA-bound and searching states could be fitted[37]. Based on calibration data (Supplementary Fig. S16e), and correcting for localization uncertainties, unbiased diffusion coefficients, and populations of the three involved states were finally derived and compared with the ground truth (see "Methods") (Fig. 5d). Interestingly, it can be seen that the retrieved diffusion coefficients for the population of searching NAPs are

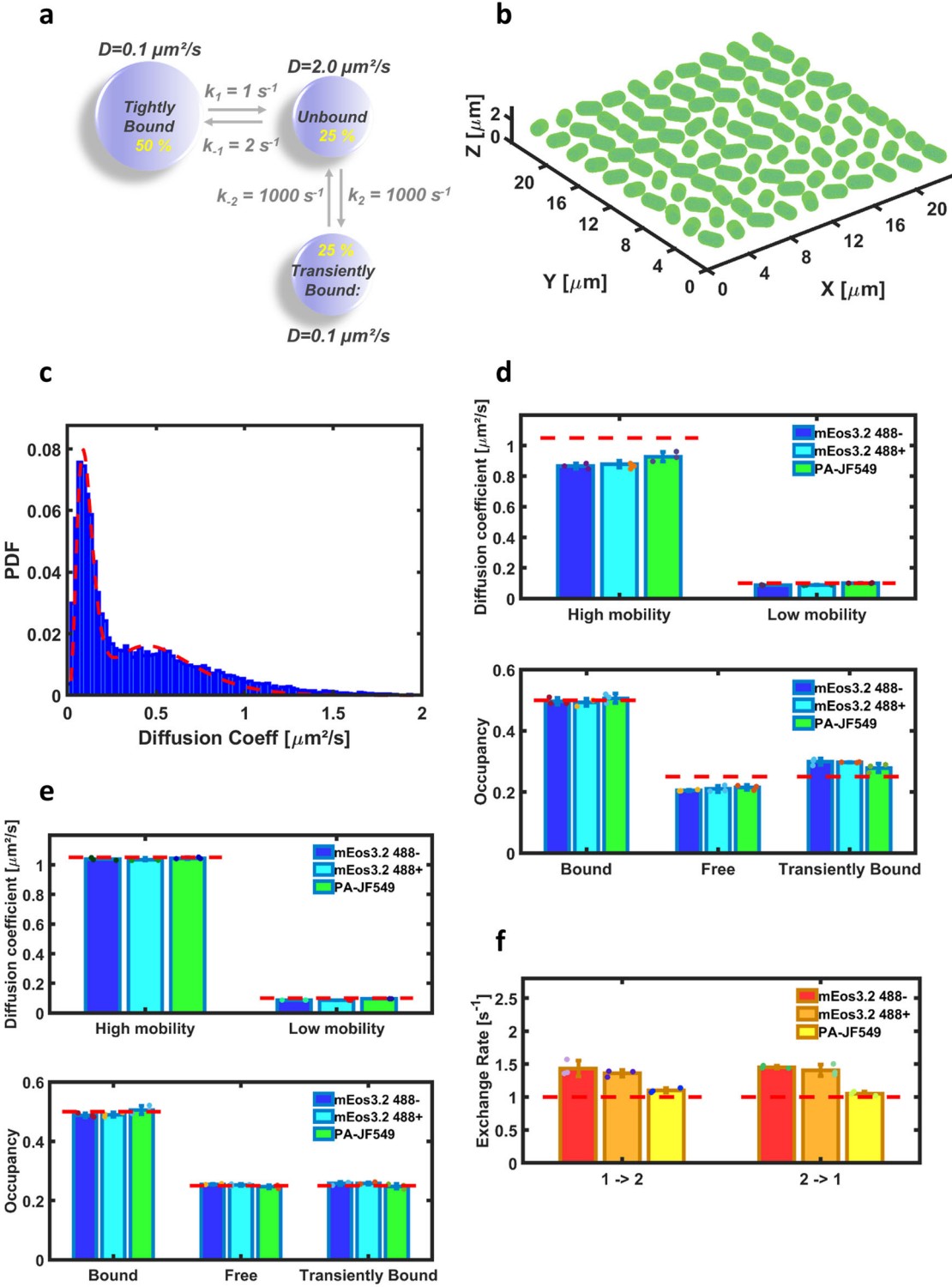

**Fig. 5 sptPALM SMIS simulations of mEos3.2- or PA-JF549-labeled NAPs diffusing within the nucleoid of E. coli bacteria. a** Employed model of NAP dynamics. **b** 3D view of the bacterial field used as the virtual SMIS sample. **c** Histogram of apparent diffusion coefficients from a mEos3.2 sample calculated from mean jump distances using 5-frames subtracks (blue), and fitted 2-state model (red). **d** Unbiased diffusion coefficients (upper panel) and diffusion state's occupancies (lower panel) recovered from MJD's-derived diffusion coefficients for mEos3.2 in the absence (blue) or presence (cyan) of 488 nm light, and for PA-JF549 (green). Ground truth values are shown as dashed red lines. **e** Unbiased diffusion coefficients (upper panel) and diffusion state's occupancies (lower panel) recovered from vbSPT-derived diffusion coefficients. Color coding as in (**d**). **f** Slow exchange rates recovered from vbSPT, for mEos3.2 in the absence (red) or presence (orange) of 488 nm light, and for PA-JF549 (yellow). 1 → 2: Tight DNA-bound state to searching state; 2 → 1: Searching state to tight DNA-bound state. Error bars for SMIS data show standard deviation for $n = 3$ simulations. Colored dots show individual data points.

underestimated relative to the ground truth, although the effect is slightly less pronounced for PA-JF549. As a consequence, the relative populations of free and weakly DNA-bound molecules are also incorrectly estimated. Further simulations show that this essentially resulted from distortions of the apparent diffusion coefficient histogram (Fig. 5c) due to both kinetic exchange and confinement (Supplementary Fig. S18).

Next we attempted to extract the slow exchange rates between the tightly DNA-bound and the searching states using vbSPT[38]. Imposing a 2-state model in vbSPT allowed accurate retrieving of unbiased diffusion coefficients and populations (Fig. 5e). Interestingly, the accuracy of the recovered exchange rates (Fig. 5f) was slightly better for mEos3.2 in the presence of 488 nm light and clearly improved for PA-JF549, showing the advantage of longer tracks even though the number of tracks was lower in PA-JF549 datasets as compared to mEos3.2 datasets. However when the number of diffusing states was not specified in vbSPT, a false 3-state model was found (Supplementary Fig. S19a), with exchange rates departing from the ground truth values, suggesting that vbSPT was biased by the strong confinement. Further simulations in which exchange kinetics were suppressed confirmed that this was indeed the case (Supplementary Fig. S20a). To attempt overcoming the 3-state-model bias due to confinement, SMIS simulations were repeated with a 5 ms total frametime (instead of 15 ms). However, a 3-state-model bias was still observed, but this time due to the fast exchange process (Supplementary Note 5, Supplementary Figs. S19b, S20–S22). Overall, the results suggest that collecting real data with varying frametimes is highly advisable: indeed SMIS data collected at 5 and 15 ms frametimes only gave consistent results if the 2-state model was chosen, strongly arguing in favor of such model.

The application above shows that SMIS can be a useful tool to investigate intertwining effects between complex diffusion dynamics, data acquisition scheme, and fluorophore photophysics in single-particle-tracking experiments.

## Discussion

The 5 examples presented above demonstrate that SMIS is able to simulate a wide variety of single-molecule imaging scenarios. Subtle behaviors can be highlighted, that are not straightforward to extract from real experiments but may yet strongly bias their proper interpretation. Many other scenarios can be simulated by SMIS. For example, combining the capabilities to incorporate both photophysical and diffusional properties of fluorophores, virtual pulse-chase or Point Accumulation for Imaging in Nanoscale Topography (PAINT) experiments can be set up. The case of a pulse-chase experiment on a neuronal cell labeled with mEos4b is shown in Supplementary Fig. S23. A PAINT experiments with the fluorogenic marker NileRed transiently binding receptor clusters on the surface of a virtual HeLa cell, using Highly Inclined and Laminated Optical sheet (HILO) illumination, is presented in Supplementary Fig. S24. Advanced simulations focused on Förster Resonance Energy Transfer (FRET) between pairs of fluorophores can also be performed. The case of a photochromic single-molecule FRET experiment between EYFP and rsTagRFP is demonstrated in Supplementary Fig. S25.

Whereas many useful SMLM simulators have been published in recent years, the simulations presented here all required specific SMIS features. A comparison of the main properties of published simulation software's is presented in Supplementary Table S15. Clearly, all packages have their advantages and drawbacks. For example, the newly introduced Virtual SMLM[9] is meant to be a real-time simulator where users can change parameters on the fly. Various experimental PSFs can also be used, which are more realistic than the ideal Gaussian or astigmatic

PSFs used in SMIS. On the other hand, fluorophores with their specific spectroscopic and photophysical signatures cannot be input in Virtual SMLM and SPT simulations cannot be performed. In fact, both SMIS and Virtual SMLM offer paradigm-shifting properties and complement each other. Supplementary Table S16 summarizes all microscope- and fluorophore-related properties that can be simulated with SMIS.

In total, simulations dealing with 12 different fluorophores, either fluorescent proteins or organic dyes, have been presented in this work. An essential property of SMIS is the possibility for users to incorporate new fluorophores according to specific experiments to be simulated. In doing so, the SMIS fluorophore database can be enriched. Presently, 28 fluorophores are defined, listed in Supplementary Table S17. Those fluorophores can be used by users willing to e.g., test different sets of imaging parameters before engaging into real experiments. Furthermore, SMIS users may modify existing fluorophores as new experimental knowledge is gathered on them. For example, photoblueing mechanisms in organic dyes[39] (e.g., photoconversion of Cy5 to Cy3) or nonlinear green-state photobleaching in EosFP variants[40] could be readily implemented.

As of today, limited photophysical knowledge is available for a majority of fluorophores, including many popular ones. More-over, the photophysical behavior of organic dyes, and -to a lesser extent- of fluorescent proteins, is heavily dependent on environmental conditions. In view of these limitations, defining meaningful photophysical models in SMIS can be challenging. Inaccurate models could even lead to misleading simulations and erroneous conclusions. Thus, SMIS must be used with care. However, several points should be noted. Firstly, for the few fluorophores that have been extensively characterized, such as fluorescent proteins of the mEos family, SMIS can be very informative in predicting non-intuitive behavior, as shown in case studies #1 and #2. Second, SMIS opens up a general framework to test models and compare simulated results with experimental data, as shown in case study #3. In this way, SMIS may contribute to foster new research on fluorophore photophysics, and should constitute a tool of growing usefulness as photophysical knowledge will increase. Third, many SMIS simulations can be performed that do not focus on photophysical models, but rather take advantage of fluorophore spectral characteristics or diffusional behavior, as shown in case studies #4 and #5. Spectral data from a huge range of fluorophores is available from Web resources such as www.fpbase.org for fluorescent proteins or https://www.thermofisher.com/order/fluorescence-spectraviewer/ for organic dyes, facilitating the introduction of new fluorophores in the SMIS database. For such simulations, simplified ad-hoc photophysical models of the fluorophores are generally sufficient, which may not be real but reproduce experimental behavior reasonably well. For example, in case study #4, such ad-hoc models have been used for the cyanine dyes CF660C and CF680, the photophysical characteristics of which have not been published. Fourth, to cope with the environmental sensitivity of fluorophores, different models for a single fluorophore can be defined based on a common set of photophysical states. For example, 4 models of Atto647N have been set up in the SMIS fluorophore database (Supplementary Table S17), where the classical 5-state model[41] (fluorescent state, triplet state, anionic radical, cationic radical, bleached state) is implemented with different interconversion rates, reproducing the experimentally observed behavior in PBS, GLOX, STORM, and ROXS buffers.

Although many advanced properties of fluorophores can be defined in SMIS, some others, as of today, cannot. This is for example the case of energy transfer mechanisms that have been shown to occur between closely separated fluorophores[42,43]. Unfortunately, accurate simulations of such mechanisms will

remain out of reach until a quantitative theoretical foundation becomes available. On the other hand, it has been shown that labels in strong proximity, for example on antibodies, tend to behave like single emitters in dSTORM conditions[44]. In such case, the approximation of a single fluorophore, as done in SMIS, may reasonably hold. Beyond, SMIS can simulate FRET, a feature not available in any published SMLM simulator to date. This constitutes a first step into modeling proximity effects.

In conclusion, beyond its pedagogical value, SMIS represents a valuable tool to help the proper setting and/or interpretation of advanced single-molecule fluorescence imaging experiments in widefield mode. The software is expected to be usable by a wide range of users, to test biological imaging experiments, evaluate photophysical models of fluorophores, or probe imaging schemes aimed at improving the power of SMLM. Thus, a benefit of SMIS could be to spare precious experimental time. SMIS also highlights that current knowledge of many fluorophore's photophysical properties is still limited, and can serve to stimulate experimental work to leverage this knowledge. SMIS could also be used to generate training datasets for machine learning algorithms.

## Methods

**Fluorophore's definition and photophysical states.** Fluorophores are defined in a SMIS database and can be created or modified by the user in an interactive manner using the "Define new fluorophore" tool. Four types of photophysical states can be specified: fluorescent states, photoactive dark states, non-photoactive dark states, and photobleached states. For each fluorescent state, excitation and fluorescence emission spectra are defined, typically from experimental data available in FPbase (www.fpbase.org) for fluorescent proteins or using e.g., SpectraViewer (www.thermofisher.com/order/fluorescence-spectraviewer) for organic fluorophores, in forms of Excel spreadsheets. Absorption spectra from photoactive dark states also need to be specified. If experimental spectra are not available, a tool is available in SMIS to generate tentative model spectra in silico. Extinction coefficients at reference wavelengths and fluorescence quantum yields of the fluorescent states are also defined. As is often the case for fluorescent proteins, a fluorescent state (typically the anionic form of a FP chromophore) can be specified to be in a pH-dependent rapid equilibrium with a dark state (typically the corresponding neutral form of the FP chromophore), according to given pKa and Hill coefficients.

Fluorophores can be set to freely tumble (representative of e.g., a live cell), or to adopt a defined orientation (representative of e.g., a frozen or fixed cell), in which case photon absorption and emission will depend on the chosen polarization of the lasers and microscope objective properties, respectively. Fluorophores can be specified to be fluorogenic (see below the section on virtual sample and fluorophore labeling). Their maturation level or labeling efficiency can be set, for example, to account for the often non-fully efficient maturation of fluorescent proteins.

**Photophysical transition matrix.** Within SMIS, transition matrices for thermally-induced and light-induced transformations between photophysical states are defined for each fluorophore, and can be manipulated by the user in an interactive manner to modify the behavior of a fluorophore or to define new fluorophores. Thermally-induced transformations are defined with rates $k_{ij}$, whereas light induced transformations are defined with quantum yields $q_{ij}$.

At each step of the data collection (ie for each frame or inter-frame), the rate of photon absorption $k_i^{hv}$ by each photoactive state of the fluorophore and due to all defined lasers is defined as:

$$k_i^{hv} = \sum_j k_i^{hv_j} \quad (1)$$

Where $k_i^{hv_j}$ is the excitation rate by laser j at the position of the fluorophore, as defined below.

Let's consider a sampling time step $\Delta t$ associated to a sampling rate $S = 1/\Delta t$, and with $\Delta t \ll T$ where T is the imaging frame time or the interframe time.

From the knowledge of the matrices $\{k_{ij}\}$ and $\{q_{ij}\}$, a global transition probability matrix $\{p_{ij}\}$ is then calculated at each step of the data collection in the following manner:

First, for a thermal process, the probability $p(k)$ of a transition at rate $k$ to occur during $\Delta t$ is given by $p(k) = k\Delta t = k/S$. Note that for a meaningful transition probability to be obtained, the property $S \gg k$ must be fulfilled, that is, a sufficiently high sampling frequency must be chosen (see below).

For a photo-induced process, let's consider $N = \Delta t/k^{hv}$, the number of photons absorbed by the fluorophore during $\Delta t$. As $N$ can be very high, to release the requirements on the used sampling frequency S, we compute the probability of a transition occurring during $\Delta t$ as:

$$p(q) = q \times \left(1 + (1-q)^1 + (1-q)^2 + \ldots + (1-q)^{N-1}\right) = 1 - (1-q)^N \quad (2)$$

Note that if $N<1$ (case of low photon absorption rate), the formula above still holds.

Finally for a transformation process which might be either thermally- or photo-induced, the transition probability is given by

$$p = p(q) + p(k) - p(k) \times p(q) \quad (3)$$

From the knowledge of the matrix $\{p_{ij}\}$, the highest probability $p_{Max}$ is determined and it is checked whether the criterium $p_{Max} < \propto$ is fulfilled where $\propto$ relates to the oversampling factor specified in SMIS (and set by default to a value of 10, ie $p_{Max} < 0.1$). If the criterium is not satisfied, the initially specified sampling frequency S is raised and the $\{p_{ij}\}$ matrix recalculated until success.

Once the frame- or interframe-specific transition probability matrix is obtained, calculation of the fluorophore photophysical trace is performed as follows: based on random number generation, occurrences along time of transitions from states $\{s_i\}$ to states $\{s_j\}$ are recorded and the first transition from the starting photophysical state $s_1$, if any, is selected, that converts $s_1$ to $s_2$ at time $t_1$. Then, the first transition from $s_2$ occurring at $t > t_1$, if any, is selected, that converts $s_2$ to $s_3$ at time $t_2$. The process is repeated until the end of the frame- or interframe-period, or until photobleaching occurs.

Whenever a fluorophore is found in one or several fluorescence states $\{s_i^F\}$ for certain times $\tau_i$ during a frametime the number of photons it emits is calculated based on computed excitation rates $k_l$ by all lasers (see below) that are on during the current frame, and the corresponding fluorescence quantum yields $Q_i$.

$$N = P\left(\sum_i \left\{Q_i \times \tau_i \times \sum_l k_l\right\}\right) \quad (4)$$

where $P$ denotes the Poissonian operator.

**Virtual sample and fluorophore labeling.** A virtual sample needs to be loaded for every fluorophore involved in the simulation. Virtual samples are typically segmented 2D images (in .tif format) or 3D stacks of images (in Matlab format), in which different parts of the sample (called "subpatterns" in SMIS) are addressed by specific pixel values. This allows to distribute fractions of the fluorophore ensemble to the individual subpatterns. Linkage errors can be specified. When fluorophores are defined to be fluorogenic, a level of fluorogenicity can be defined for each sub pattern, a useful feature to simulate e.g PAINT experiments with fluorogenic chromophores. A set of synthetic virtual samples (e.g., NPCs, HeLa cell, E. coli bacteria …) can be created by the SMIS "Create virtual samples" tool.

**Computation of single-molecule diffusion trajectories.** Single fluorophores may diffuse in 2D or 3D throughout their associated virtual sample, according to specified diffusion coefficients and/or velocities (in case of directed motion). Typically, a fluorophore diffuses in a confined manner within the subpattern it belongs to. However, fluorophores can be set to switch between several diffusion regimes within a specific subpattern, or to exchange between subpatterns according to a transition matrix defined by the user. This feature allows simulating complex diffusion scenarios, with fluorophores binding and unbinding from specific sample features at defined rates. Importantly, SMIS offers the option to calculate diffusion with sub-frame resolution, which provides motion-blurred fluorescence emission spots on the detector, proper temporal averaging and a more faithful description of the explored space, notably in the case of rapid motion.

The SMIS single-molecule diffusion modeling capabilities open the possibility to simulate complex sptPALM experiments, as well as ensemble techniques such as FRAP or pulse-chase experiments. PAINT experiments can also be simulated where fluorophores can transiently bind to their target to provide sparse single-molecule fluorescence emission.

**Computation of excitation rates.** The excitation rate of a rapidly tumbling molecule located at position $(x, y)$ is given by: $k = \varepsilon P \lambda \frac{(10^{-6})Ln(10)}{\mathcal{N}hc}$ where $\varepsilon$ is the extinction coefficient of the considered photophysical state at the excitation wavelength $\lambda$, $P(x, y)$ is the laser power-density at $(x, y)$, $\mathcal{N}$ is the Avogadro number, $h$ is the Planck constant and $c$ is the speed of light[45].

For a fixed oriented molecule with spherical coordinates $\theta$ and $\varphi$, the excitation rate of the absorbance dipole is angle dependent. For a circularly polarized laser beam whose electromagnetic field is assumed to be parallel to the objective focal plane, the excitation rate is given by $k_\theta = \frac{3}{2} k \cos^2 \theta$, where $\theta$ is measured from the focal plane. For a linearly polarized laser beam whose electromagnetic field is in the objective focal plane and aligned with the x axis, the excitation rate becomes $k_{\theta,\varphi} = 3 k \cos^2 \theta \cos^2 \varphi$ (ref. [45]).

**Virtual microscope description.** The experimental setup is simulated by a number of parameters describing laser illumination, microscope optical properties and detector characteristics.

Any number of laser beams with Gaussian or flat spatial profiles within the focal plane can be defined, with chosen wavelengths, powers, polarization (circular or linear) and FWHM (full width at half maximum). 3D illumination can be simulated in Widefield, TIRF or HILO configurations. Complex laser sequence patterns can be generated during data collection, such as for example pre-bleaching steps, alternate excitation, and ramping-up or Fermi profile (typically for activation lasers).

The microscope is described by an objective of given numerical aperture NA, which is assumed to give rise to a Gaussian point spread function (PSF) with standard full width at half maximum (FWHM = $1.22\lambda$/NA where $\lambda$ is the emission wavelength). The photon collection efficiency takes into account the orientation of the emitting dipoles[46].

Transmission efficiencies of inserted emission or dichroic filters (possibly multi-bands), as well as overall transmission efficiency of the optical setup are also taken into account. Data from commercial filters can be input.

The EMCCD detector is described by the effective pixel size, dark current noise, readout noise and gain (counts/detected photon)[3].

SMLM data sets with a defined number of frames and frametime and inter-frametime can then be generated. For each single-molecule emission, the spatial (throughout the microscope PSF) and spectral (throughout the emission spectrum) distributions of emitted photons are calculated based on Poissonian statistics.

A level of fluorescence background, optionally proportional to the local illumination power density, can be defined. Autofluorescence can be set to vary spatially according to given input patterns, and temporally according to given decay rates.

The program outputs the stack of acquisition frames (.tif), diffraction limited images (.tif), laser illumination profiles (.tif), and for each molecule, its $x$ and $y$ coordinates, fluorescence emission trace, number of detected photons, eventual diffusion trace, amongst other properties (.mat).

SMIS includes possibilities to simulate dual-channel detection, multi-color experiments (with $N$ different fluorophores), sample drift, single-molecule Förster resonance energy transfer (FRET) between two fluorophores, and localized activation/bleaching similar to what can be normally achieved with a Fluorescence Recovery After Photobleaching (FRAP) module.

**SMIS performance**. SMIS was developed under Matlab 2022a (https://fr.mathworks.com). The SMIS Graphical User Interface was developed with Matlab App Designer (Supplementary Fig. S26). Although SMIS is not meant to behave as a real-time simulation software[9], it performs reasonably fast (running time of ~1 s/frame for a typical 3D SMLM experiment involving full photophysics of 3200 mEos4b molecules decorating 100 NPCs on a 2015 high-grade PC (HP Z-book 15 G2 equipped with a i7-4810MQ processor). Complex scenarios may take more time. However, SMIS offers parallel computing to improve performance on multi-CPU computers. Future developments including C++ code will further improve the speed performance of SMIS.

As for any simulation software, the produced results merely reflect the input models, and accuracy needs to be traded to some extent for speed. SMIS mainly focuses on the impact of fluorophore properties, and thus, although it is able to simulate a complex virtual microscope environment, it is not designed to provide the most accurate optical response, e.g., it doesn't use for example vectorial or experimental PSFs. Although SMIS is fully compatible with 3D imaging, only the astigmatism-based 3D localization method is currently implemented.

**Processing of SMIS data generated in the described case studies**. SMIS data generated in Fig. 1 were localized, filtered and merged with Thunderstorm[47]. Image rendering used ImageJ (https://imagej.nih.gov/ij/)[48].

SMIS data generated in Fig. 2 were localized, filtered and merged with Thunderstorm[47]. Coordinate-based colocalization and nearest neighbor distance measurements used the CBC tool of Thunderstorm.

SMIS data generated in Fig. 3 were processed with SMAP[49] using identical parameters as those employed in Diekmann et al.[11].

SMIS data generated in Fig. 4 were localized and filtered with Thunderstorm[47]. Unmixing was realized with SpliVisu[28]. Crosstalk analysis was based on nearest neighbor calculation between measured and ground truth localizations of the different fluorophores, using the CBC tool of Thunderstorm. A crosstalk event between a pair of fluorophores A and B was assigned if the nearest ground truth localization from a measured localization of fluorophore B was that of a fluorophore A and the distance was less than 5 times the measured localization precision. Image rendering used ImageJ.

SMIS data generated in Fig. 5 were processed with Trackit[35], using either the nearest neighbor track search option, or the more advanced U-Track-based option[36]. Exchange rates were extracted with vbSPT[38]. No blinking gap was allowed in the reconstituted tracks, as imposed by the two used evaluation procedures (histograms of diffusion coefficients based on Mean Jump Distances (MJDs) and vbSPT). Fitting of the MJDs-based diffusion coefficient histograms exactly followed the procedures described in Stracy et al.[31]. Retrieval of the fractional population's p1, p2 and p3 of tightly DNA-bound, weakly DNA-bound and freely diffusing molecules was based on the following: first, fitting of the MJD-based diffusion coefficient histograms or vbSPT results, assuming a 2-state model in both cases, provided the relative populations p1 and p23 of tightly DNA-bound

and searching molecules, as well as their associated apparent diffusion coefficients $D_{app,1}$ and $D_{app,23}$. Those apparent diffusion coefficients were first corrected by subtracting the contribution of localization error ($\sigma^2/\Delta t$, where $\Delta t$ is the frame time and $\sigma$ is the median localization uncertainty over the considered dataset (~25 nm and ~12 nm for mEos3.2 and PA-JF549 data, respectively, estimated with Thunderstorm)). Then, unbiased diffusion coefficients $D_1$ and $D_{23}$ were extracted based on the calibration curves of Supplementary Fig. S16e. The fractional population $p$ of weakly DNA-bound molecules relative to all searching molecules was given by:

$$D_{23} = p \times (D_1) + (1 - p) \times (D_{free}) \tag{5}$$

where $D_{free} = 2.0\ \mu m^2$/s was assumed to be extracted, as described in Stracy et al.[31], from calibrated measurements in perturbed cells devoid of nucleoid where NAPs are freely diffusing. Finally $p_2 = p \times p_{23}$ and $p_3 = (1 - p) \times p_{23}$.

All figures were generated using ImageJ (https://imagej.nih.gov/ij/) or with homemade routines written in Matlab.

**Statistics and reproducibility**. For all SMIS applications described in this paper, results extracted from processed SMIS-generated image stacks were generally derived from $n = 3$ simulations carried out for all conditions tested. Whenever relevant, data from single simulations were split in $n = 3$ subsets to extract standard deviations. Standard deviations were calculated using the std Matlab command. Ground truth data and calibration data shown in the paper were generally derived from $n = 1$ simulation for all conditions tested. SMIS simulations do not reproduce biological variability and are only affected by variability in fluorophore labeling, stochastic changes in photophysical states or diffusional behavior of the fluorophores, Poissonian noise and detector noise. They are thus generally highly reproducible.

**Reporting summary**. Further information on research design is available in the Nature Portfolio Reporting Summary linked to this article.

## Code availability

The source code, Matlab app, as well as freely usable standalone versions of SMIS vsn2.1 for Windows can be downloaded at https://mts-commsbio.nature.com/ (Supplementary Software). Versions of SMIS vsn2.1 for Windows, Linux and Mac OS are available on GitHub at https://github.com/DominiqueBourgeois/SMIS, or can be downloaded via Zenodo (https://doi.org/10.5281/zenodo.7503996)

## Data availability

Simulated datasets generated in this work and corresponding SMIS simulation input files are available from the author on reasonable request. Example SMIS simulation input files for all case studies presented in this article are included in the SMIS example's library provided with the software (Supplementary Software). Source data underlying the graphs presented in the main and supplementary figures of this manuscript can be downloaded at https://mts-commsbio.nature.com/ (Supplementary Data 1).

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

## Acknowledgements

I am indebted to the many people that helped me throughout this work thanks to inspiring discussions, in particular Leonid Andronov, Mark Bates, Ulrike Endesfelder, Viktorija Glembockyte, Juliette Griffié, Florian Levet, Martin Linden, Jonas Ries, Daniel Sage, Philip Tinnefeld and Jochem Vink. I also thank my lab mates for multiple discussions and for testing SMIS and providing suggestions, in particular Jip Wulffele. The SMIS GUI was essentially developed during Covid confinement, but the virus is not acknowledged. This work was supported by the Agence Nationale de la Recherche (grants no. ANR-17-CE11-0047-01 and ANR-20-CE11-0013-01).

## Competing interests

The author declares no competing interests.
