## [Peer Review File · Communications Biology]

Reviewers' comments:

Reviewer #2 (Remarks to the Author):

The article titled 'Single Molecule Imaging Simulations (SMIS) with Full Fluorophore Photophysics' authored by Dominique Bourgeois, is about simulating the complex photophysics of fluorescent molecules and the influence of the imaging environment on them while performing fluorescence microscopy. The research paper is particularly interesting for scientists working with single-molecule fluorescence microscope experiments. However, I think the author needs to compare and contrast the results from his SMIS tool with the other existing tools. My major comments and minor corrections on the paper are as follows:

Major comments:

- A major concern while reading this article is that the author failed to provide the experimental validation of the results observed from the four example simulations by the SMIS tool. The author brushes it slightly in example 3 for Cy5 photophysics, but for other examples, the author only compares results with the ground-truth data used for the simulations. This creates an important apprehension about the novelty of applying the simulation results to real-world experiments. For instance, in example 2, SMIS demonstrates the crosstalk between Dendra2 and mEos4b. Can it be validated in experiments? Without such validations, it is hard to accept the novelty of this article.
- It is unclear from the article how widespread this tool can be applied beyond the examples reported here. The author claims the tool can be expanded for any fluorophore beyond the examples given (Main text 2nd paragraph, end sentence). However, the author has not given clear insights on the procedure for simulating a new fluorophore and what properties of it are required for the tool.
- Many of the graphs in figures 1 & 3 are confusing, mainly because the legends are missing in the graphs (e.g., Figure 1 panel C, Figure 3 panel E & F). The captions detail it, but it would help the reader interpret the results better when proper legends are shown in the panels.
- As seen in Figure 3F and detailed in the supplementary information, the SMIS fails to corroborate with the experimental data of dSTORM on Cy5 dye. The author has only detailed the discrepancy results but has not discussed them well in the paper. It requires more revision in this part of the paper. Also, I would like to know whether the author has attempted to add the new possible long-lived dark state in the SMIS tool, and the results of it might add novelty to the paper. If it is not attempted, the author should provide the reasons.

Minor comments:

- I think the paper's title is too strong, especially the term 'Full Fluorophore Photophysics.' The author only provides four different examples of how the tool simulates some of the biophysical properties of a few fluorophores. But I find the title a strong statement that may mislead the reader to think the tool will simulate all the fluorophore properties, especially when there are still many unknowns in fluorophore photophysics (The author even mentions this limitation in conclusion). Hence, it must be re-worded appropriately.
- Some sentences are quite long, which makes the paper less readable.
- In the abstract, the last sentence is not clear.

There is no experimental validation of the simulation. Also, several simulation platforms are available, including Virtual-SMLM, "a virtual environment for real-time interactive SMLM acquisition" (bioRxiv, <https://www.biorxiv.org/content/10.1101/2020.03.05.967893v1.full>) and the author failed to explain why SMLS is better than Virtual SMLM. This manuscript might be suitable for any mathematical or photophysical journals.

Reviewer #3 (Remarks to the Author):

The manuscript by Dr. Bourgeois introduces a simulation software for single-molecule fluorescence imaging that accounts for photophysical states and transitions of fluorophores.

This very rigorous work showcases how the complexity of fluorophore photophysics impacts

different types of single-molecule imaging experiments. The reasons are the population of various energy states in fluorophores and transitions between these, which are influenced by experimental parameters such as excitation wavelength, buffer pH/redox potential, and the local chemical environment. Knowledge of the photophysical landscape allows maneuvering the experiment such that 'best-possible' results are obtained, i.e., as close as possible to the ground truth.

The examples cover main applications in single-molecule imaging. A limitation I see is that photophysical models are available for very few fluorophores only, e.g., the photoswitchable fluorophores mEos, Dendra and Alexa647. Given the large number of fluorophores used in single-molecule imaging, how would the author suggest to use this software for fluorophores that are not that well characterized? Can this information be extracted from accessible sources? This would be particularly helpful given the suggestion to implement this strategy in biology research.

Recent work has shown how the change of photophysical properties at close distances impacts single-molecule imaging (PMID: 35915194), which requires a multi-molecule energy model. This is a very common scenario if multi-labeled antibodies are used, e.g., in dSTORM, as well as for FRET at short distances. Can the author comment on how this could be incorporated in the simulation approach?

The photophysics of organic dyes is, in general, probably very hard to tackle with a simulation: unprotected from the environment, organic fluorophores are influenced by the molecular environment (e.g., aromatic amino acids, embedding in protein tags, DNA nucleotides etc.), the molecular orientation, and other effects such as 'photo-blueing' (PMID: 33633409). What would be a pragmatic solution in such situations?

In my opinion, the author did a great job in showcasing how sub-optimal imaging conditions can impact the quality of single-molecule imaging results. The examples are chosen well, and cover a broad spectrum of applications. What I would like to encourage the authors is to elaborate pragmatic solutions that can be adopted by the broad community of users with the so different backgrounds.

The software was not available to me for testing, which would have been nice to evaluate how simpler photophysical schemes would alter the results (e.g., a 4-state model as compared to a 10/12-state model). However, the point I want to make is different: from what I understand from the article, the software is written in MatLab, which will be a barrier for users because of the need to pay for licenses. The single-molecule field has had such experience several times in the past: even the availability of a compiled version (which would not need the MatLab license) is of limited help if compatibility issues arise that require editing the source code and re-compilation.

A few minor points:

Does the simulation software account for linkage errors? This is not strictly needed for the simulation of photophysics.

SMLM experiments also need to be set up with respect to the local 'dimensionality' of a structure (e.g., point clusters, or linear filaments, or dense 2D/3D structures) and density. This does need matching imaging conditions (see also the above comment on fluorophore density and altered photophysics). It was not entirely clear to me whether for the Nup96 example, with four overlaying fluorescent proteins at a single site, the local activation probability is responsible for the lower quality of some images.

How was the localization precision calculated?

Page 2, "virtually labelled in 2D", please rephrase to specify that the 2D images were generated from an NPC model with 32 Nup96-mEos.

Figure 4, I found the sequence of panels in the figure not intuitive to follow, yet that might be quite subjective.

A few original methods and tools are mentioned that may be referenced, e.g. dSTORM, SPT-PALM, CBC, Fiji; as much, earlier work reporting that photoswitching rates need to match

Reviewer #2 (Remarks to the Author):

The article titled 'Single Molecule Imaging Simulations (SMIS) with Full Fluorophore Photophysics' authored by Dominique Bourgeois, is about simulating the complex photophysics of fluorescent molecules and the influence of the imaging environment on them while performing fluorescence microscopy. The research paper is particularly interesting for scientists working with single-molecule fluorescence microscope experiments. However, I think the author needs to compare and contrast the results from his SMIS tool with the other existing tools. My major comments and minor corrections on the paper are as follows:

Major comments:

- A major concern while reading this article is that the author failed to provide the experimental validation of the results observed from the four example simulations by the SMIS tool. The author brushes it slightly in example 3 for Cy5 photophysics, but for other examples, the author only compares results with the ground-truth data used for the simulations. This creates an important apprehension about the novelty of applying the simulation results to real-world experiments. For instance, in example 2, SMIS demonstrates the crosstalk between Dendra2 and mEos4b. Can it be validated in experiments? Without such validations, it is hard to accept the novelty of this article.

I think this criticism stems from a confusion which directly results from a lack of clarity in my original manuscript. Experimental data would validate (or not) the fluorophore photophysical models input in SMIS, but not the SMIS simulator itself. Gathering experimental data for each case-study would turn the paper into a very wide investigation of fluorophore photophysics, rather than introducing what I believe is a paradigm-changing simulator. This would change the scope of the paper.

Studying fluorophore photophysics is definitely a key asset of SMIS, and this is what is done in example #3. Here simulated data is compared with published experimental data with the aim to investigate Cy5 photophysics. The results allow concluding that the input Cy5 model is not able to fully account for experimentally observed NPC-Nup96-Alexa647 data (Alexa647 is a close analog to Cy5). The possibility to pinpoint such discrepancy, to my opinion, is a strength of SMIS.

*It cannot be envisaged in the context of this paper to do similar investigations for the 3 other case-studies. Each one would require several months (or even years ?) of experimental work with very diverse expertise. For example, in case study #1, one would need to produce a NPC-Nup96-mEos4b CRISPR-Cas9 cell line, and master optimized imaging of NPCs; in case study # 2 suitable fusion constructs and a microscope adapted to primed-photoconversion should be available; and in case study # 4 (# 5 in the revised version), a strong expertise with nucleoid associated proteins in *E. coli* would be required ... My lab is definitely interested in all these topics, but they all represent quite major efforts. I believe our future experimental research will benefit from SMIS and vice versa.*

This being said, I thank the reviewer for pointing out this confusion. I reworded the manuscript so that it now clearly states (i) that the results presented in the various case studies are only meant to illustrate the versatility and complexity of experiments that can be simulated with SMIS, (ii) that the conclusions drawn would be valid only as long as the input photophysical models of the used fluorophores are correct, and (iii) that SMIS opens up possibilities to examine and potentially refine those photophysical models, as exemplified in case study #3.

- It is unclear from the article how widespread this tool can be applied beyond the examples reported here. The author claims the tool can be expanded for any fluorophore beyond the examples given (Main text 2nd paragraph, end sentence). However, the author has not given clear insights on the procedure for simulating a new fluorophore and what properties of it are required for the tool.

Please see also my detailed answer to reviewer # 3. A more condensed answer is provided below.

SMIS is not limited concerning the number of fluorophores it can deal with. In the submitted manuscript, simulations were performed with 9 different fluorophores. Technically, any fluorophore can be defined in SMIS by the users, as long as spectral data are available (from e.g. fpbase.org (fluorescent proteins) or other sources such as the ThermoFisher SpectralViewer (organic dyes)), and with variable level of refinement concerning dark or photobleached states, depending on the goal of the simulations. This is described at the beginning of the methods section, which I modified to improve clarity.

This being said, it is a fact that there is a lack of knowledge concerning the complex photophysics of many fluorophores, notably for organic dyes that, in addition, can behave very differently depending on environmental conditions. Whereas SMIS can contribute improving this knowledge, by allowing refinement of photophysical models to make simulations match experimental data (like case study # 3), it can also be used in a more “basic” way: simple “ad hoc” photophysical models can be set up, that are not real, or are largely incomplete, but allow producing still realistic simulated data. In many cases this is enough to take advantage of the fact that SMIS can deal with the full spectroscopic signature of the fluorophores. This allows for example to simulate crosstalk or FRET efficiencies in multicolor experiments. This aspect of SMIS was not sufficiently highlighted in the submitted version of the manuscript.

Overall, I have revised the paper in the following manner: (i) I have added a 5th case-study (now # 4) to the paper dealing with spectral demixing for multicolor experiments, in which photophysics of the employed dyes (AF647, CF680, CF660C, AF700) is not the focus. (ii) I have added a Supplementary Table, listing the fluorophores that are currently defined in the SMIS database. (iii) I have added discussion at the end of the manuscript to address the concern of limited fundamental photophysical knowledge, and highlight how SMIS can be employed by users with little knowledge of fluorophores.

Overall, the reviewer’s comment has been extremely helpful to substantially improve my manuscript in a way that will hopefully convince the readers that many fluorophores can indeed be simulated in SMIS.

- Many of the graphs in figures 1 & 3 are confusing, mainly because the legends are missing in the graphs (e.g., Figure 1 panel C, Figure 3 panel E & F). The captions detail it, but it would help the reader interpret the results better when proper legends are shown in the panels.

I thank the reviewer for this suggestion, and added extra legends on the specified graphs.

- As seen in Figure 3F and detailed in the supplementary information, the SMIS fails to corroborate with the experimental data of dSTORM on Cy5 dye. The author has only detailed the discrepancy results but has not discussed them well in the paper. It requires more revision in this part of the paper. Also, I would like to know whether the author has attempted to add the new possible long-lived dark state in the SMIS tool, and the results of it might add novelty to the paper. If it is not attempted, the author should provide the reasons.

I respectfully disagree with the reviewer. SMIS does not “fail to corroborate with the experimental data of dSTORM on Cy5 dye”, it rather suggests that the photophysical model of Gidi et al, although it is a very elaborate model, is still insufficient to account for experimentally observed dSTORM data. As detailed in the manuscript, the Gidi et al model can be partially tuned (by changing rates) to better approach experimental results, which can be putatively justified by slight chemical differences between Alexa647 and Cy5, and by the fact that these dyes are used in different environments (Halo-tag for STORM data, in solution for Gidi et al). However, the study also suggests that the Gidi et al model lacks a ms-lived dark state, and is thus likely incomplete. The study already goes quite deep in photophysics, which is almost beyond the scope of the paper. It is the result of 6 months of investigations involving many discussions with specialists in the field (who are acknowledged at the end of the paper), some of which being main authors of the Gidi et al paper and of the Diekmann et al paper. So it is unclear to me what the reviewer means by “not discussed them well in the paper”. A quite thorough discussion is presented in Supplementary Text 3 and Supplementary Note 1, which are already too specialized to be inserted in the main manuscript. I really feel that going further would turn the paper into a very deep photochemistry-oriented paper. The true relevance of the putative missing long-lived dark state, although it is hinted by the cited references 20 and 21 (numbers from the submitted manuscript), would now need to be established by comprehensive experimental photochemical studies. This would represent a major effort, in the frame of collaborative work, and the results would likely deserve a separate publication, possibly in a high profile chemical journal. The overall message is that SMIS can trigger such future investigations, and facilitate them.

However, to cope with the reviewer’s concern, I have done the following: I have tried to improve the clarity of the text for case-study #3. Also, in the new case study dealing with spectral demixing, I have considered Alexa647 and the other employed STORM dyes with a simplified photophysical model that does incorporate a ms-lived dark state as the precursor to the sulfur-adduct dark state. The model is “ad hoc”, ie has no established photochemical foundation and is oversimplified, but does behave as expected under dSTORM conditions.

Minor comments:

- I think the paper's title is too strong, especially the term 'Full Fluorophore Photophysics.' The author only provides four different examples of how the tool simulates some of the biophysical properties of a few fluorophores. But I find the title a strong statement that may mislead the reader to think the tool will simulate all the fluorophore properties, especially when there are still many unknowns in fluorophore photophysics (The author even mentions this limitation in conclusion). Hence, it must be re-worded appropriately.

Although SMIS incorporates many more fluorophore properties and offers more flexibility, to my knowledge, than is possible with previous simulation software, the reviewer is fully right: there remains a lot of unknown in fluorophore photophysics that cannot be accurately modeled. Therefore, I replaced the term “Full Fluorophore Photophysics” in the title by “Advanced Fluorophore Photophysics”.

- Some sentences are quite long, which makes the paper less readable.

Wherever possible I tried to split or simplify such sentences.

- In the abstract, the last sentence is not clear.

I splitted the sentence into 2, and rewrote it to better highlight the broad capabilities of SMIS.

There is no experimental validation of the simulation. Also, several simulation platforms are available, including Virtual-SMLM, "a virtual environment for real-time interactive SMLM acquisition" (bioRxiv, <https://www.biorxiv.org/content/10.1101/2020.03.05.967893v1.full>) and the author failed to explain why SMIS is better than Virtual SMLM.

The reviewer is fully right that not enough comparison with available single-molecule simulators was discussed in the submitted version of the paper. To remedy to this, I incorporated Supplementary Table S16 that presents the most important aspects of published simulation software, and I reemphasize in the discussion section the new possibilities opened by SMIS. Furthermore, I added Supplementary Table S17 that summarizes all microscope and fluorophore properties that can be simulated with SMIS.

Differences between Virtual SMLM and SMIS are in fact quite profound. The sentence mentioning Virtual SMLM in the introduction of the submitted paper was potentially misleading (being in fact "too generous" with Virtual SMLM) and has been modified. While Virtual SMLM is essentially meant to be a real-time simulator (where users can change parameters, notably laser intensities, on the fly), specific fluorophores with their defined spectroscopic signatures and photophysical descriptions cannot be input, so that, for example, wavelength-dependent responses and crosstalk effects cannot be studied. Another very important difference is that Virtual SMLM does not allow incorporating the diffusional behavior of the fluorophores, preventing for example sptPALM, PAINT, FRAP or pulse-chase experiments to be performed. In fact, I believe none of the simulations shown in the paper can be done with Virtual SMLM (nor any available similar simulator). Overall, whereas SMIS is not meant to be real time, it enables to simulate more diverse and more complex experiments using properly defined fluorophores. On the other hand, Virtual SMLM is most likely simpler to use, and offers more possibilities for 3D PSF's, notably the use of experimental PSF's (which I intend to implement in SMIS in a future release). In total, the two software's are highly complementary. Of note, a thorough benchmarking with Virtual SMLM is not possible because as of today this software has not been released.

This manuscript might be suitable for any mathematical or photophysical journals.

I really hope that this rebuttal letter and the revised version of the manuscript will now convince the reviewer that SMIS opens up novel avenues for single-molecule imaging simulations, of potential interest to a wide audience.

Reviewer #3 (Remarks to the Author):

The manuscript by Dr. Bourgeois introduces a simulation software for single-molecule fluorescence imaging that accounts for photophysical states and transitions of fluorophores.

This very rigorous work showcases how the complexity of fluorophore photophysics impacts different types of single-molecule imaging experiments. The reasons are the population of various energy states in fluorophores and transitions between these, which are influenced by experimental parameters such as excitation wavelength, buffer pH/redox potential, and the local chemical environment. Knowledge of the photophysical landscape allows maneuvering the

experiment such that 'best-possible' results are obtained, i.e., as close as possible to the ground truth.

The examples cover main applications in single-molecule imaging. A limitation I see is that photophysical models are available for very few fluorophores only, e.g., the photoswitchable fluorophores mEos, Dendra and Alexa647. Given the large number of fluorophores used in single-molecule imaging, how would the author suggest to use this software for fluorophores that are not that well characterized? Can this information be extracted from accessible sources? This would be particularly helpful given the suggestion to implement this strategy in biology research.

The limitation the reviewer is pointing at can be understood in two ways: it can be understood as a limitation of SMIS itself, or as a limitation due to the lack of photophysical knowledge on fluorophores in general.

In view of the first interpretation, SMIS is not limited concerning the number of fluorophores it can deal with. In the submitted manuscript, simulations were performed with mEos3.2, mEos4b, Dendra2, PAmCherry, Cy5, PA-JF549 (in the main manuscript) but also with NileRed, EYFP and rsTagRFP (in the supporting information, figures S24 and S25 (revised manuscript numbers)), a total of 9 different fluorophores. Importantly, any fluorophore can be defined in SMIS by the users, using spectral data from e.g. fpbase.org (fluorescent proteins) or other sources such as the ThermoFisher SpectralViewer (organic dyes), and with variable level of refinement concerning dark or photobleached states, depending on the goal of the simulations. This is described at the beginning of the methods section, which I modified to improve clarity.

In view of the second interpretation, the reviewer is fully right that there is a lack of knowledge concerning the complex photophysics of many fluorophores, notably for organic dyes that, in addition, can behave very differently depending on environmental conditions. With this in mind, there are two ways SMIS can be used:

- ⇒ SMIS can contribute improving this knowledge, by allowing refinement of photophysical models to make simulations match experimental data. This is the focus of the case-study # 3, but could also apply to case studies # 1 and # 2. In the case of photobleaching mentioned (below) by the reviewer, it would be straightforward to implement this mechanism into e.g. cyanine photophysical models (photoconversion from Cy5 to Cy3), and for example simulate the predicted crosstalk artifacts in multicolor experiments, depending on applied illumination conditions, and compare with experimental data.*
- ⇒ SMIS can also be used in a more "basic" way, ie employing simple "ad hoc" photophysical models that are not real, or are largely incomplete, but allow producing relatively realistic simulated data. However, a major advantage of SMIS remains here as compared to other simulators: SMIS can deal with the spectroscopic signature of the fluorophores, so that simulated data are sensitive to laser wavelengths and provide accurate crosstalk or FRET efficiencies in multicolor experiments. This aspect of SMIS was not sufficiently highlighted in the submitted version of the manuscript.*

Overall, I have revised the paper in the following manner: (i) I have added a 5th case-study (now # 4) to the paper dealing with spectral demixing for multicolor experiments. Here, the photophysics of the employed dyes (AF647, CF680, CF660C, AF700) are not the focus (ad-hoc models for STORM imaging are used), but rather the spectral shift between the fluorophores is of importance. Together with the

case-study on single-particle-tracking (now # 5), and with the 3 examples provided in supplementary figures S23-25 (which do not involve complex photophysics), I hope this will provide a more precise view of the capabilities of SMIS in simulating a large variety of experiments, even when limited photophysical knowledge is available. (ii) I have added Supplementary Table S15, listing the fluorophores that are currently defined in the SMIS database. 28 fluorophores are listed, although again it should be clear that defining more fluorophores, or refining the photophysical models of those fluorophore, can be done with a dedicated tool in SMIS. (iii) I have added substantial discussion at the end of the manuscript to precisely address the concern of limited fundamental photophysical knowledge, and highlight how SMIS can be employed by users with little knowledge of fluorophores. In particular, I hope this will convince the readers that it should be useful and easy for non-expert users to use SMIS, as, much like in real experiments, they only need to choose fluorophores and enter imaging parameters (e.g. laser wavelengths and power) to perform simulations. Instead, with currently available simulators, sets of rather arbitrary interconversion rates between states, with little link to reality, have to be input.

Overall, the reviewer's comment has been extremely helpful to improve my manuscript in a way that will hopefully convince the readers that many fluorophores can be simulated in SMIS.

Recent work has shown how the change of photophysical properties at close distances impacts single-molecule imaging (PMID: 35915194), which requires a multi-molecule energy model. This is a very common scenario if multi-labeled antibodies are used, e.g., in dSTORM, as well as for FRET at short distances. Can the author comment on how this could be incorporated in the simulation approach?

The reviewer is fully right that intricate energy transfer mechanisms may occur between very closely separated fluorophores. This is elegantly demonstrated, and taken advantage of, in the mentioned publication by the Sauer group. A similar effect is likely at play between Cy3 and Cy5 in the original version of STORM microscopy (Bates et al. PRL 94, 108101, 2005). As, to my knowledge, there is no quantitative theoretical foundation for such energy transfer, SMIS is indeed unable to simulate them. On the other hand, it has also been shown that multiple-labeled antibodies tend to behave like single emitters in STORM conditions (Helmerich et al. ACS Nano 2020, 14, 10, 12629–12641), in which case the approximation of a single fluorophore may reasonably hold.

Beyond, SMIS can simulate FRET, which to my knowledge is not available in any published single-molecule simulators to date and constitutes a first but significant step into modeling proximity effects. FRET in SMIS only deals with rapidly tumbling fluorophores ($\kappa^2=2/3$) but takes into account transitioning between photophysical states. An example was given in the submitted version of the paper (FRET with RSFPs, now Supplementary Figure S25), which probably was not sufficiently highlighted. In the revised version of the manuscript, discussion based on the reviewer's comment is added, where the benefit and limitations of SMIS regarding the modeling of energy transfer effects are more clearly mentioned.

The photophysics of organic dyes is, in general, probably very hard to tackle with a simulation: unprotected from the environment, organic fluorophores are influenced by the molecular environment (e.g., aromatic amino acids, embedding in protein tags, DNA nucleotides etc.), the molecular orientation, and other effects such as 'photo-blueing' (PMID: 33633409). What would be a pragmatic solution in such situations?

The reviewer is fully correct. Due to this sensitivity to environmental conditions, it will probably never be possible to derive a complete photophysical model of an organic dye, at least in terms of interconversion rates between states. The highly complex model of Cy5 described in the case study # 3 is tuned for specific STORM buffer conditions. In general, although there is hardly any limit in the complexity of photophysical models that can be technically defined in SMIS, only models tuned to correspond to specific physicochemical conditions should be reasonably used in practice. This was not sufficiently well highlighted in the submitted manuscript. Furthermore, three points are important to note:

- ⇒ As discussed above for the case of photo-blueing, novel discoveries in the photophysical behavior of fluorophores can in principle be implemented in SMIS and the effects tested e.g. as a function of illumination conditions.*
- ⇒ Different models can be derived based on a common set of photophysical states, by tuning interconversion rates. The case of Atto647N is given in the revised manuscript, where the classical 5-state model (fluorescent state, triplet state, anionic radical, cationic radical, bleached state, Ha & Tinnefeld Annu. Rev. Phys. Chem. 2012. 63:595–617) can be implemented with different rates, resulting in various behaviors in PBS, GLOX, STORM and ROXS buffers.*
- ⇒ Simplified ad-hoc models can be implemented for the many fluorophores for which no description of their photophysical behavior is available and for simulations not specifically focusing on photophysics. This is now highlighted in the newly introduced case study #4.*

Finally, SMIS can simulate to some extent the effects of fluorophore dipole orientation: laser polarization can be defined (as linear or circular) and fluorophores can be set to tumble or to adopt a fixed orientation, or possibly to jump randomly to a new orientation at a defined rate.

All these points are now detailed in the discussion section of the revised manuscript.

In my opinion, the author did a great job in showcasing how sub-optimal imaging conditions can impact the quality of single-molecule imaging results. The examples are chosen well, and cover a broad spectrum of applications. What I would like to encourage the authors is to elaborate pragmatic solutions that can be adopted by the broad community of users with the so different backgrounds.

I thank the reviewer for this appreciation. I hope that, thanks to the comments of both reviewer's, I have managed in the revised version of the manuscript to show that SMIS can effectively be used by a broad community of users. I acknowledge that the submitted version looked somewhat obscure, as in fact the presented examples were all quite complex, and the paper did not make it clear enough that a greater variety of more simple simulations with diverse fluorophores can be performed. The added case-study # 4 on spectral demixing and the added discussion will hopefully now convince the readers that SMIS can, for example, be of significant help to design new experiments (choice of fluorophores, imaging conditions, etc ...) before actually conducting it.

The software was not available to me for testing, which would have been nice to evaluate how simpler photophysical schemes would alter the results (e.g., a 4-state model as compared to a 10/12-state model). However, the point I want to make is different: from what I understand from the article, the software is written in MatLab, which will be a barrier for users because of the need

to pay for licenses. The single-molecule field has had such experience several times in the past: even the availability of a compiled version (which would not need the MatLab license) is of limited help if compatibility issues arise that require editing the source code and re-compilation.

Concerning the 1st point raised by the reviewer: in some cases, the reviewer is right: simpler models may reproduce the behavior from more complex models reasonably well. This is for example the case in the (new) case-study # 4, where ad-hoc 4-state models for Alexa647, CF660C and CF680 are used instead of the much more complex model of Cy5 in case study #3. However, a major goal of SMIS is to provide a general framework to allow reproducing the genuine photophysical states of fluorophores, which is useful in many cases. For example, it would be impossible to reproduce the results from the investigated case-studies # 1 or 2, where 10-12 state models are used, if those models would be simplified to a much lower number of states (not mentioning the importance of the spectral signatures of all states). Establishing pH-sensitivity of photoconvertible fluorescent proteins requires already 4 states (anionic and protonated states for both the green and the red fluorophore), blinking in the red and green states, 2 additional dark states, bleaching, 1 additional state, and whenever relevant, intersystem crossing, 1 or 2 additional states.

Now about the code: a few days after submission, I had communicated to the editor a link to download a freely executable version the software. I apologize that this was probably done too late, so that the reviewers probably remained unaware of this.

*Community members have different opinions about adequate computer languages to use. I admit that Matlab-based standalone executables may not run on every platform depending on system architecture, but definitely no license is required. Other open source languages may pose system compatibility issues as well. Matlab continues to be very widely used throughout the single molecule localization microscopy community, and Matlab codes have been published in high-impact Nature journals recently (see for example Ries, J. SMAP: a modular super-resolution microscopy analysis platform for SMLM data. Nat Methods **17**, 870–872 (2020)). One advantage of Matlab is that those users experienced with it and willing to modify or upgrade the SMIS code will be able to do so. This being said, I tend to agree with the reviewer and if I would start this work today, I would probably use an open-source language such as Python. I am sure the reviewer will understand that it is impossible for me to rapidly transfer 10 years of Matlab code development into another language.*

In the revised version of the manuscript I now provide standalone (freely executable) versions of SMIS for Windows, Linux and Mac OS. In addition, the source code and a Matlab app are also available. This should guarantee sufficiently broad access to the simulator. The SMIS software can be downloaded at <https://github.com/DominiqueBourgeois/SMIS>.

A few minor points:

Does the simulation software account for linkage errors? This is not strictly needed for the simulation of photophysics.

As the reviewer points out, this is not critical for most simulations that can benefit from SMIS. Nevertheless, based on the reviewer's comment, I added a linkage error option in the current version of SMIS. Of note, using a linkage error is only meaningful if the digital resolution of the employed

virtual sample is significantly smaller (ie better) than the linkage error. SMIS automatically controls whether this is the case or not, and warns the user if necessary.

SMLM experiments also need to be set up with respect to the local 'dimensionality' of a structure (e.g., point clusters, or linear filaments, or dense 2D/3D structures) and density. This does need matching imaging conditions (see also the above comment on fluorophore density and altered photophysics). It was not entirely clear to me whether for the Nup96 example, with four overlaying fluorescent proteins at a single site, the local activation probability is responsible for the lower quality of some images.

SMIS is exactly designed to answer this sort of questions: the imaging conditions are given as on a real microscope (e.g. readout and activation laser power densities), with virtual samples exhibiting their own "local dimensionality", labeled with realistic fluorophores. From the produced image stacks and the knowledge of the ground truth, it is then possible to figure out what is causing defects in the rendered images. In case-study #1, it is suggested that image quality results from a complex balance between activation probability (which produce spot overlaps if it is too high, but leaves molecule being bleached in the green state before activation if it is too low), and readout intensity (which bleaches green molecules if it is too high, but enhances spot overlaps due to prolonged on times and reduces the photon budget per localization if it is too low).

How was the localization precision calculated?

Image stacks were processed like experimental data, as specified in the Methods section (Processing of SMIS data). Thus, localization precision was in most cases derived from the Thunderstorm software (except for case study #3, where the SMAP software was used).

Page 2, "virtually labelled in 2D", please rephrase to specify that the 2D images were generated from an NPC model with 32 Nup96-mEos.

Done, thank you for the suggestion.

Figure 4, I found the sequence of panels in the figure not intuitive to follow, yet that might be quite subjective.

The reviewer is right, it is in fact difficult to find an ideal arrangement for all the panels in this figure because 2 of the panels contain double graphics. Hopefully the revised layout, which shows the panels in a more intuitive sequential order, will be suitable.

A few original methods and tools are mentioned that may be referenced, e.g. dSTORM, SPT-PALM, CBC, Fiji; as much, earlier work reporting that photoswitching rates need to match

I apologize for this omission. The mentioned methods are now properly referenced.

REVIEWERS' COMMENTS:

Reviewer #2 (Remarks to the Author):

The author has addressed all the concerns listed in my previous review. The free version of the software is online and accessible to test and use. The manuscript is now in good shape for the wider community who wants to study fluorophore photophysics.